# Sarcoeleganolides C–G, Five New Cembranes from the South China Sea Soft Coral *Sarcophyton elegans*

**DOI:** 10.3390/md20090574

**Published:** 2022-09-10

**Authors:** Cili Wang, Jiarui Zhang, Xing Shi, Kai Li, Fengling Li, Xuli Tang, Guoqiang Li, Pinglin Li

**Affiliations:** 1Key Laboratory of Marine Drugs, Chinese Ministry of Education, School of Medicine and Pharmacy, Ocean University of China, Qingdao 266003, China; 2Laboratory of Marine Drugs and Biological Products, National Laboratory for Marine Science and Technology, Qingdao 266235, China; 3Biology Institute, Qilu University of Technology (Shandong Academy of Sciences), Jinan 250103, China; 4College of Chemistry and Chemical Engineering, Ocean University of China, Qingdao 266100, China

**Keywords:** *Sarcophyton elegans*, sarcoeleganolides C–G, cembranes, anti-inflammation activity

## Abstract

Five new cembranes, named sarcoeleganolides C–G (**1**–**5**), along with three known analogs (**6**–**8**) were isolated from soft coral *Sarcophyton elegans* collected from the Yagong Island, South China Sea. Their structures and absolute configurations were determined by extensive spectroscopic analysis, QM-NMR, and TDDFT-ECD calculations. In addition, compound **3** exhibited better anti-inflammation activity compared to the indomethacin as a positive control in zebrafish at 20 μM.

## 1. Introduction

Soft corals have been recognized as a rich source of nature products with diverse chemical structures. Soft corals of the genus *Sarcophyton* (family Alcyoniidae) are widely regarded as an important source of cembranoids [1,2,3,4,5,6,7,8]. These marine secondary metabolites are featured by a 14-membered carbocyclic ring [3], and showed a broad spectrum of biological activities, such as anti-inflammatory [9], cytotoxic [10], antibacterial [11], antifouling [12], neuroprotective activities [13]. Due to their complex structures and multiple bioactivities, the level of interest in cembranoids from *Sarcophyton* soft corals has continued to grow over the years, and impressive achievements have been made. In previous studies, numbers of cembranoids such as sarcomililate A [14], 13-oxo-thunbergol [11], ximaoglaucumins A–F [15], ximaolides H–L [16], and trocheliophols A–S [17] were isolated from Sar*cophyton* soft corals.

Their fascinating structures and extensive biological activities make them attractive for further investigation. To pursue novel metabolites with bioactivities, a continuous search of the soft coral *Sarcophyton elegans* collected from the Yagong Island in the South China Sea led to the discovery of five new cembranoids, named sarcoeleganolides C–G (**1**–**5**), along with three known analogs, trocheliolide B (**6**) [18], (−)-sartrochine (**7**) [19], and 7α-hydroxy-Δ^8(19)^-deepoxysarcophine (**8**) [20], as shown in Figure 1. Herein, the isolation, structure elucidation and biological activity of these isolated compounds are reported.

## 2. Results

Sarcoeleganolide C (**1**), which was isolated as a colorless oil, gave a molecular formula of C_20_H_28_O_3_ by its HRESIMS ion peak at *m*/*z* 317.2114 [M + H]^+^, implying seven degrees of unsaturation. The 1D NMR data (Table 1) and HSQC spectrum of **1** revealed the presence of 20 carbons belonging to four methyls (three olefinic, and one sp^3^ hybridized), six methylenes (all sp^3^ hybridized), four methines (two olefinic and two oxygenated), and six quaternary carbons (four olefinic, one sp^3^ hybridize, and one carbonyl). These data indicate that compound **1** was a cembrane-type diterpenoid.

The planar framework of **1** was elucidated by ^1^H-^1^H COSY and HMBC spectra (Figure 2). Four spin systems were established by the ^1^H-^1^H COSY correlations from H-2 to H-3; H-5 to H-7; H-9 to H-11, and H-13 to H-14. As previously reported, 3, 4-epoxy-cembranolides [21,22], a trisubstituted epoxide ring located at C-3 and C-4, were deduced by the downfield chemical shift of C-3 (*δ*_C_ 61.5) and C-4 (*δ*_C_ 61.5) and HMBC correlations from H_3_-18 to C-3, C-4, and C-5. Based on the above data, together with the key HMBC correlation from H_3_-19 to C-7, C-8, and C-9; H_3_-20 to C-11, C-12, and C-13; H_3_-17 to C-1, C-15, and C-16; H-14a (*δ*_H_ 2.74) to C-1, C-2, and C-15 the connection of the carbon skeleton was permitted. Thus, compound **1** was deduced as a cembranoid possessing a trisubstituted epoxide. In the NOESY spectrum of **1** (Figure 3), the correlations of H_3_-19/H-6a (*δ*_H_ 2.20), H_3_-20/H-10a (*δ*_H_ 2.26) indicate that the Δ^7^ and Δ^11^ double bonds could be of an *E*-configuration. The NOESY correlation of H-2/H_3_-18 indicates that these protons were on the same side. In addition, considering the geometry of the 3-(*E*)-olefin in co-isolates, the epoxide of **1** should be in an anti-relationship between H-3 and H_3_-18, which was further confirmed by the ^13^C NMR chemical shift calculation for the DP4^+^ calculations (Appendix A) [23]. Finally, the absolute configurations of **1** were defined as 2*S*, 3*R*, and 4*R* by TDDFT-ECD calculations (Figure 4).

Sarcoeleganolide D (**2**), a colorless oil, had a molecular formula of C_21_H_30_O_4_ on the basis of its HRESIMS ion peak at *m*/*z* 347.2221 [M + H]^+^, requiring seven degrees of unsaturation. The ^1^H and ^13^C NMR data of **2** (Table 1) resemble that of (−)-sartrochine (**7**), a known cembranoid previously isolated from the soft coral *Sarcophyton trochliphroum*. In fact, the structure of **2** was truly similar to **7**, with the exception of a methoxyl at C-2 in **2** instead of the proton in **7**. This deduction was further proven by the HMBC correlation (Figure 2) from the H_3_-21 (*δ*_H_ 3.14) to C-2, along with the significant downfield shift observed for C-2 (*δ*_C_ 108.3). Then, the relative configurations of **2** were deduced on the basis of the NOESY experiment (Figure 3). The NOESY correlations of H-3/H_2_-5 (*δ*_H_ 2.20 and *δ*_H_ 2.20), and H_3_-19/H-6a (*δ*_H_ 2.35) established the *E* geometry of the Δ^3^ and Δ^7^ double bonds. The NOESY correlations of H_3_-21/H-13a (*δ*_H_ 1.68), and H-11/13a (*δ*_H_ 1.68) indicate that these protons were all co-facial. Moreover, the NOESY correlation of H_3_-20/H-13b (*δ*_H_ 1.89) suggests these protons were on the opposite side. Finally, the absolute configuration of **2** was defined by TDDFT-ECD calculations (Figure 4).

Sarcoeleganolide E (**3**), a colorless oil, possessed the molecular formula C_22_H_30_O_5_, as indicated by its HRESIMS ion peak at *m*/*z* 397.1991 [M + Na]^+^. The comparison of the 1D NMR data (Table 1) of **3** and **6** indicate similarities between them. The 2D NMR data of **3** (Appendix A and Figure 2) indicate the plane structure was identical to **6**, suggesting that **3** should be a stereoisomer of **6**. The relative configurations of **3** were deduced by the NOESY spectrum (Figure 3). By the NOESY correlation of H_3_-19/H-7, the geometry of the Δ^7^ double bonds was assigned to be a *Z*-configuration, which was further confirmed by the downfield chemical shift of C-19 (*δ*_C_ 22.4), revealing the major difference in configurations between **3** and **6**. The *E* geometry of the Δ^3^ double bonds was established by the observed NOESY correlations of H-3/H-5a (*δ*_H_ 2.38) and H-2/H_3_-18. Based on the above data, the NOESY correlations of H_3_-18/H-5b (*δ*_H_ 2.04) indicate the inverse orientation of H-2 and H-3, which was further confirmed by the coupling constants (*J*_2,3_ = 10.0 Hz). The diagnostic NOESY correlations of H-13a (*δ*_H_ 1.09)/H-11, and H-13a/H-2 assigned H-11 and H-2 were all co-facial. The NOESY correlations of H-6/H-3, H-6/H-9a (*δ*_H_ 2.61), and H_3_-20/H-9a suggest that H_3_-20, H-3, and H-6 were on the same side of the ring system. Hence, the relative configurations of **3** were deduced, and finally, the absolute configurations of **3** were defined by TDDFT-ECD calculations (Figure 4).

Sarcoeleganolide F (**4**) was obtained as a colorless oil. The HRESIMS ion peak at *m*/*z* 369.2034 [M + Na]^+^ suggests the molecular formula was C_21_H_30_O_4_, suggestive of seven degrees of unsaturation. The ^1^H NMR spectrum (Table 1) and HSQC spectrum confirm the presence of 21 carbons, five methyls (three olefinic, one oxygenated, and one sp^3^ hybridized), five methylenes (all sp^3^ hybridized), four methines (three olefinic and one oxygenated), and seven quaternary carbons (five olefinic, one oxygenated, and one carbonyl). By analysis of these data above, compound **4** was speculated to be a cembrane nucleus.

The carbon skeleton of **4** was established by the ^1^H-^1^H COSY and HMBC experiments (Figure 2). The separate spin systems of H-5/H-6/H-7, H-9/H-10/H-11, and H-13/H-14 were established by the ^1^H-^1^H COSY correlations. The HMBC correlations from H_3_-21 (*δ*_H_ 3.21) to C-6 (*δ*_C_ 73.4) indicate the presence of a methoxy group located at C-6. A trisubstituted double bond located at C-2 and C-3 was proven by HMBC correlations from H-3 (*δ*_H_ 5.23) to C-2 (*δ*_H_ 148.2). Combined with the significant HMBC correlations from H_3_-17 to C-1, C-15, and C-16; H_3_-18 to C-3, C-4, and C-5; H_3_-19 to C-7, C-8, and C-9; H_3_-20 to C-11, C-12, and C-13; and H_2_-14 to C-1, C-2, and C-15, the planar framework of **4** was established. The relative configurations of **4** were deduced by the NOESY spectrum (Figure 3). The strong NOESY cross-peaks of H_3_-19/H-6 (*δ*_H_ 4.30) and H-11/H_2_-13 (*δ*_H_ 2.33) established the *E* geometries of the Δ^7^ and Δ^11^ double bonds, and the NOESY cross-peaks of H-3/H_2_-14 (*δ*_H_ 2.54) established the *Z* geometry of the Δ^2^ double bonds. By the ^13^C NMR chemical shift calculations for the DP4^+^ calculations (Appendix A), the configurations were defined as 4*R** and 6*S**, which were further confirmed by the NOESY correlations of H_3_-18/H-5a (*δ*_H_ 2.16) and H-6/H-5a. Finally, the absolute configurations of **4** were defined by TDDFT-ECD calculations (Figure 4).

Sarcoeleganolide G (**5**) was isolated as a colorless oil with a molecular formula of C_20_H_28_O_3_, established by the HRESIMS ion peak at *m*/*z* 339.1928 [M + Na]^+^. A survey of the literature revealed that the 1D NMR data of compound **5** (Table 1) were similar to those of compound **8**, a known cembrane diterpenoid isolated from the Red Sea soft coral *Sarcophyton glaucum*. In fact, compound **5** had the same functional groups as **8**, except for the migration of the Δ^8^ double bonds in **8** to the Δ^12^ double bonds in **5**, and the hydroxy group at the C-7 position in **8** to C-11 in **5**. These variations of the functional groups were further proven by the HMBC correlations from H_3_-20 to C-11, C-12, and C-13, and from H_3_-19 to C-7, C-8 and C-9. Furthermore, other detailed HMBC correlations and ^1^H-^1^H COSY correlations helped complete the planar framework of **5** (Figure 2). In the NOESY spectrum of **5** (Figure 3), the correlations of H-3/H-5a (*δ*_H_ 2.20), H-2/H_3_-18, and H-7/H_2_-9 (*δ*_H_ 2.03) indicate that the geometries of Δ^3^ and Δ^7^double bonds were of an *E*-configuration. By the NOESY correlations of H-2/H-14a (*δ*_H_ 2.26) and H-11/H-14a (*δ*_H_ 2.26), the relative configurations were defined as 2*S** and 11*S**. Finally, the absolute configurations of **5** were defined by TDDFT-ECD calculations (Figure 4).

Although the anti-inflammatory activity of cembranoids in zebrafish models has been reported previously [24], it is still not very common. Hence, we aimed to seek newer cembranoids with anti-inflammatory activity in zebrafish models. These new compounds (**1**–**5**) were evaluated for anti-inflammatory activity in CuSO4-induced transgenic fluorescent zebrafish. CuSO_4_ can produce an intense acute inflammatory response in the neuromast and mechanosensorial cells in the lateral line of zebrafish, stimulating the infiltration of macrophages [25,26,27]. Then the number of macrophages surrounding the neuromast in the zebrafish was observed and imaged under a fluorescence microscope (Appendix A, Section 3). The results are shown in Figure 5. In CuSO_4_-induced transgenic fluorescent zebrafish, compound **3** could alleviate migration and decreased the number of macrophages surrounding the neuromast in the zebrafish, showing stronger anti-inflammatory activity than the indomethacin, which was used as the positive control at 20 μM, while other compounds showed no anti-inflammatory activity, as shown in Figure 5. 

## 3. Materials and Methods

### 3.1. General Experimental Procedures

Optical rotations were measured on a Jasco P-1020 digital polarimeter (Jasco, Tokyo, Japan). The UV spectra were recorded on a Beckman DU640 spectrophotometer (Beckman Ltd., Shanghai, China). The CD spectra were obtained on a Jasco J-810 spectropolarimeter (Jasco, Tokyo, Japan). The NMR spectra were measured by Agilent 500 MHz (Agilent, Beijing, China), JEOL JNMECP 600 spectrometers (JEOL, Beijing, China). The 7.26 ppm and 77.16 ppm resonances of CDCl_3_ were used as internal references for the ^1^H and ^13^C NMR spectra, respectively. The 7.16 ppm and 128.06 ppm resonances of C_6_D_6_ were used as internal references for the ^1^H and ^13^C NMR spectra, respectively. The HRESIMS spectra were measured on Micromass Q-Tof Ultima GLOBAL GAA076LC mass spectrometers (Autospec-Ultima-TOF, Waters, Shanghai, China). Semi-preparative HPLC was performed using a Waters 1525 pump (Waters, Singapore) equipped with a 2998 photodiode array detector and a YMC C18 column (YMC, 10 × 250 mm, 5 μm). Silica gel (200–300 mesh, 300–400 mesh, and silica gel H, Qingdao Marine Chemical Factory, Qingdao, China) was used for column chromatography.

### 3.2. Animal Material

The soft coral *Sarcophyton elegans* was collected from Xisha Island (YaGong Island) in the South China Sea in 2018 and frozen immediately after collection. The specimen was identified by Ping-Jyun Sung, at the Institute of Marine Biotechnology, the National Museum of Marine Biology and Aquarium, Pingtung 944, Taiwan. The voucher specimen (No. xs-18-yg-114) was deposited at the State Key Laboratory of Marine Drugs, Ocean University of China, People’s Republic of China.

### 3.3. Extraction and Isolation 

A frozen specimen of *Sarcophyton elegans* (7.2 kg, wet weight) was homogenized and then exhaustively extracted with CH_3_OH six times (3 days each time) at room temperature. The combined solutions were concentrated in vacuo and were then subsequently desalted by redissolving with CH_3_OH to yield a residue (178.0 g). The crude extract was subjected to silica gel vacuum column chromatography eluted with a gradient of petroleum/acetone (400:1–1:1, *v*/*v*) and subsequently eluted with a gradient of CH_2_Cl_2_/MeOH (20:1–1:1, *v*/*v*) to obtain fourteen fractions (Frs.1–Frs.14). Each fraction was detected by TLC. Frs.5 was subjected to a silica gel vacuum column chromatography (petroleum/acetone, from 100:1 to 1:1, *v*/*v*) to give three subfractions Frs.5.1–Frs.5.3. Frs.5.1 was separated by semi-preparative HPLC (ODS, 5 µm, 250 × 10 mm; MeOH/H_2_O, 70:30, *v*/*v*; 1.5 mL/min) to afford **1** (5.0 mg, t_R_ = 72 min). Frs.5.2 was separated by semi-preparative HPLC (ODS, 5 µm, 250 × 10 mm; MeOH/H_2_O, 65:35, *v*/*v*; 1.5 mL/min) to afford **2** (3.7 mg, t_R_ = 70 min). Frs.6 was subjected to silica gel vacuum column chromatography (petroleum/acetone, from 100:1 to 1:1, *v*/*v*) to give two subfractions, Frs.6.1–Frs.6.2. Frs.6.2 was separated by semi-preparative HPLC (ODS, 5 µm, 250 × 10 mm; MeOH/H_2_O, 65:35, *v*/*v*; 1.5 mL/min) to afford **4** (2.0 mg, t_R_ = 54 min) and **5** (3.5 mg, t_R_ = 27 min). Frs.7 was subjected to silica gel vacuum column chromatography (petroleum/acetone, from 50:1 to 1:1, *v*/*v*) to give six subfractions, Frs.7.1–Frs.7.6. Frs.7.4 was separated by semi-preparative HPLC (ODS, 5 µm, 250 × 10 mm; MeOH/H_2_O, 65:35, *v*/*v*; 1.5 mL/min) to afford **3** (2.0 mg, t_R_ = 48 min).

Sarcoeleganolide C (**1**): colorless oil; [α]25D +23.3 (c 1.0, MeOH); UV (MeOH) λmax (log ε) = 200 (0.91) nm; HRESIMS *m*/*z* 317.2114 [M+H]^+^ (calcd. for C_20_H_29_O_3_^+^, 317.2111). For ^1^H NMR and ^13^C NMR data, see Table 1.

Sarcoeleganolide D (**2**): colorless oil; [α]25D −36.7 (c 1.0, MeOH); UV (MeOH) λmax (log ε) = 200 (0.92) nm; HRESIMS *m*/*z* 347.2221 [M + H]^+^ (calcd. for C_21_H_31_O_4_^+^, 347.2217). For ^1^H NMR and ^13^C NMR data, see Table 1.

Sarcoeleganolide E (**3**): colorless oil; [α]25D +45.5 (c 0.5, MeOH); UV (MeOH) λmax (log ε) = 197 (2.13) nm; HRESIMS *m*/*z* 397.1991 [M + Na]^+^ (calcd. For C_22_H_30_O_5_Na^+^, 397.1985). For ^1^H NMR and ^13^C NMR data, see Table 1.

Sarcoeleganolide F (**4**): colorless oil; [α]25D +66.2 (c 0.5, MeOH); UV (MeOH) λmax (log ε) = 201 (2.25) nm, 280 (1.58) nm; HRESIMS *m*/*z* 364.2481 [M + NH_4_]^+^ (calcd. For C_21_H_34_O_4_N^+^, 364.2482) and 369.2034 [M + Na]^+^ (calcd. For C_21_H_30_O_4_Na^+^, 369.2036). For ^1^H NMR and ^13^C NMR data, see Table 1.

Sarcoeleganolide G (**5**): colorless oil; [α]25D +54.2 (c 0.5, MeOH); UV (MeOH) λmax (log ε) = 195 (0.57) nm; HRESIMS *m*/*z* 339.1928 [M + H]^+^ (calcd. for C_20_H_28_O_3_Na^+^, 339.1931). For ^1^H NMR and ^13^C NMR data, see Table 1.

### 3.4. Anti-Inflammatory Activity Assay

Healthy macrophage fluorescent transgenic zebrafish (Tg: zlyz-EGFP) was provided by the Biology Institute of the Shandong Academy of Science (Jinan, China). Zebrafish maintenance and the anti-inflammation assay were carried out as previously described [26]. Each zebrafish larva was photographed by a fluorescence microscope (AXIO, Zom.V16), and the number of macrophages around the nerve mound was calculated using Image-Pro Plus 6.0 software (Rockville, MD, USA) [28]. One-way analysis of variance was performed using GraphPad Prism 7.00 software (San Diego, CA, USA) [29]. Sarcoeleganolides C–G (**1**–**5**) were tested for anti-inflammatory activities with zebrafish models. Three days post-fertilization (dpf) healthy macrophage fluorescent transgenic zebrafish were used as animal models to evaluate the anti-inflammatory effects of **1**–**5**.

## 4. Conclusions

In our search for soft coral *Sarcophyton elegans* collected from the South China Sea, five new cembranes, named sarcoeleganolides C–G (**1**–**5**), and three known analogs, trocheliolide B (**6**), (−)-sartrochine (**7**), and 7α-hydroxy-Δ^8(19)^-deepoxysarcophine (**8**), were isolated. In addition, their structures and absolute configurations (**1**–**5**) were determined by extensive spectroscopic analysis, QM-NMR, and TDDFT-ECD calculations. Among them, compound **3** showed better anti-inflammatory activity, compared to the indomethacin as the positive control at 20 μM in the zebrafish model. This research enriches the chemical libraries of soft coral *Sarcophyton elegans* and provides a basis for developing new drugs.

## Figures and Tables

**Figure 1 marinedrugs-20-00574-f001:**
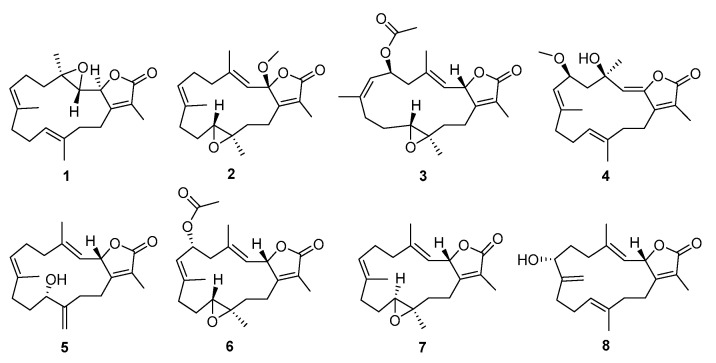
Structures of compounds **1**–**8**.

**Figure 2 marinedrugs-20-00574-f002:**
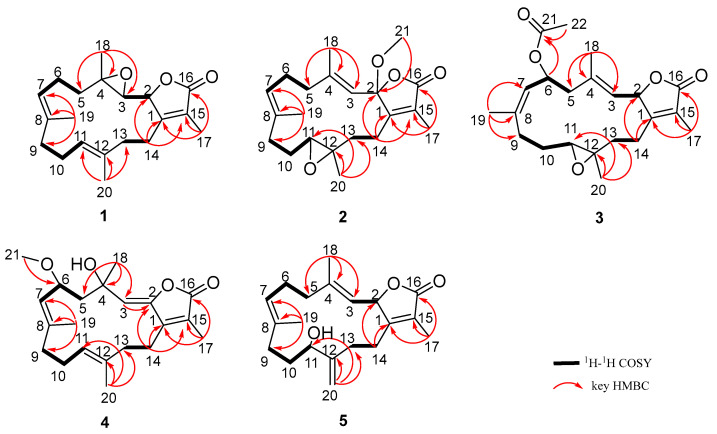
Selected ^1^H–^1^H COSY and HMBC correlations of compounds **1**–**5**.

**Figure 3 marinedrugs-20-00574-f003:**
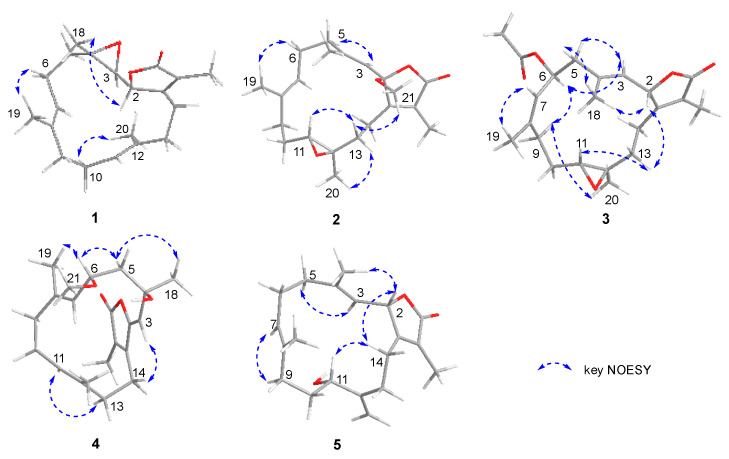
Key NOESY and 1D-NOE correlations of **1**–**5**.

**Figure 4 marinedrugs-20-00574-f004:**
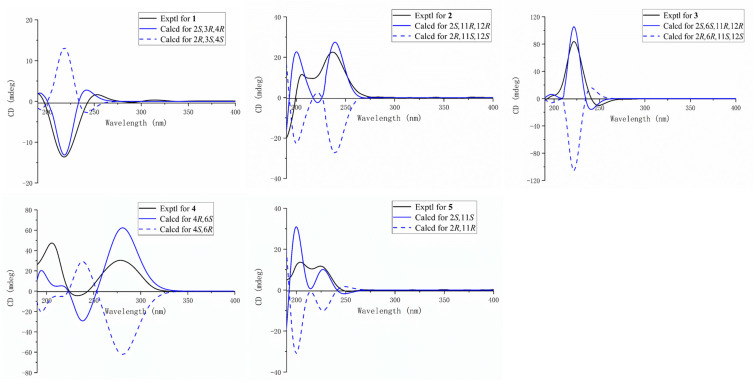
Experimental and calculated ECD spectra of **1**–**5**.

**Figure 5 marinedrugs-20-00574-f005:**
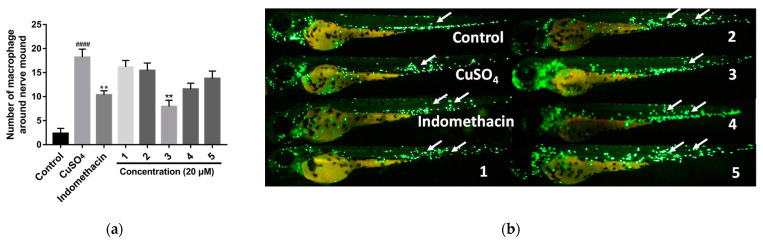
(**a**) Quantitative analysis of macrophages in the region of inflammatory sites in zebrafish treated with sarcoeleganolides C–G (**1**–**5**) in zebrafish at 20 μM. (**b**) Images of inflammatory sites in CuSO_4_-induced transgenic fluorescent zebrafish (Tg:zlyz-EGFP) expressing enhanced green fluorescent protein (EGFP) treated with sarcoeleganolides C–G (**1**–**5**), using indomethacin as a positive control. #### Indicates that the CuSO_4_ model group shows very significant differences compared to the control group (*p* < 0.01). ** Indicates that the sample groups show significant differences compared to the CuSO_4_ model group (*p* < 0.01).

**Table 1 marinedrugs-20-00574-t001:** ^1^H and ^13^C NMR data of sarcoeleganolides C–G (**1**–**5**).

No.	1 ^a^		2 ^a^		3 ^b^		4 ^a^		5 ^a^	
	*δ*H^c^ (*J* in Hz)	*δ*C ^d^	*δ*H ^c^ (*J* in Hz)	*δ*C ^d^	*δ*H ^e^ (*J* in Hz)	*δ*C ^f^	*δ*H ^e^ (*J* in Hz)	*δ*C ^f^	*δ*H ^e^ (*J* in Hz)	*δ*C ^f^
1		160.7, qC		158.3, qC		160.0, qC		151.9, qC		161.6, qC
2	4.94, m	79.1, CH		108.3, qC	4.91, d, (10.0)	78.4, CH		148.2, qC	5.45, d, (10.5)	79.9, CH
3	2.77, d, (4.2)	61.5, CH	5.16, s	120.6, CH	4.74, d, (10.0)	125.3, CH	5.23, s	117.0, CH	4.89, d, (10.5)	120.0, CH
4		61.5, qC		143.8, qC		139.0, qC		72.0, qC		144.7, qC
5a	1.37, m	38.8, CH_2_	2.20, m	40.2, CH_2_	2.38, dd, (10.0, 3.0)	46.2, CH_2_	2.16, m	48.5, CH_2_	2.32, m	39.5, CH_2_
5b	2.08, m		2.20, m		2.04, t, (11.5)		2.03, m		2.20, m	
6a	2.20, m	23.7, CH_2_	2.35, m	24.6, CH_2_	5.36, td, (10.0, 2.0)	71.1, CH	4.30, t, (9.0)	73.4, CH	2.20, m	24.2, CH_2_
6b	2.08, m		2.14, m						2.35, m	
7	5.05, t, (7.2)	124.3, CH	5.02, t, (6.6)	125.7, CH	5.19, d, (10.0)	127.3, CH	4.88, d, (9.0)	124.8, CH	4.92, d, (5.0)	123.0, CH
8		135.2, qC		134.3, qC		141.9, qC		141.1, qC		135.5, qC
9a	2.11, m	38.8, CH_2_	2.29, m	37.0, CH_2_	2.61, td, (14.0, 2.5)	29.1, CH_2_	2.10, m	38.7, CH_2_	2.03, m	33.9, CH_2_
9b	2.18, m		2.03, m		1.76, m		2.10, m		2.03, m	
10a	2.26, m	24.4, CH_2_	2.06, m	24.2, CH_2_	1.24, m	24.3, CH_2_	2.22, m	24.3, CH_2_	1.71, m	34.4, CH_2_
10b	2.20, m		1.34, m		1.86, m		2.10, m		1.71, m	
11	5.09, t, (6.6)	126.4, CH	2.69, dd, (9.6, 3.3)	61.5, CH	2.30, dd, (10.5, 2.5)	58.8, CH	4.86, d, (4.0)	125.9, CH	3.98, t, (6.5)	72.1, CH
12		133.6, qC		61.6, qC		59.9, qC		131.7, qC		151.9, qC
13a	2.04, m	36.6, CH_2_	1.68, m	34.0, CH_2_	1.09, m	35.2, CH_2_	2.33, m	36.2, CH_2_	2.24, m	32.1, CH_2_
13b	2.45, m		1.89, m		1.63, m		2.33, m		2.17, m	
14a	2.74, m	24.9, CH_2_	2.45, m	23.4, CH_2_	1.74, m	22.0, CH_2_	2.54, m	22.5, CH_2_	2.26, m	27.0, CH_2_
14b	2.39, m		2.14, m		1.59, m		2.54, m		2.46, m	
15		124.0, qC		126.4, qC		124.0, qC		123.2, qC		124.2, qC
16		174.4, qC		172.2, qC		173.9, qC		170.0, qC		174.9, qC
17	1.83, s	8.8, CH_3_	1.90, s	8.8, CH_3_	1.61, s	8.8, CH_3_	1.93, s	9.3, CH_3_	1.87, s	9.0, CH_3_
18	1.53, s	17.9, CH_3_	1.57, s	15.9, CH_3_	1.34, s	18.3, CH_3_	1.45, s	32.9, CH_3_	1.78, s	15.9, CH_3_
19	1.58, s	16.1, CH_3_	1.66, s	15.0, CH_3_	1.46, s	22.4, CH_3_	1.66, s	17.2, CH_3_	1.64, s	17.1, CH_3_
20	1.68, s	17.0, CH_3_	1.29, s	16.6, CH_3_	1.11, s	17.3, CH_3_	1.60, s	17.1, CH_3_	5.19, s; 5.02, s	110.9, CH_2_
21			3.14, s	50.2, CH_3_		169.4, qC	3.21, s	55.1, CH_3_		
22					1.65, s	20.9, CH_3_				

^a^ Spectra recorded in chloroform -d4. ^b^ Spectra recorded in benzene -d6. ^c^ Spectra recorded at 600 MHz. ^d^ Spectra recorded at 150 MHz. ^e^ Spectra recorded at 500 MHz. ^f^ Spectra recorded at 125 MHz.

## Data Availability

Data are contained within the article or Appendix A.

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
