# Peer review of "Sarcoeleganolides C–G, Five New Cembranes from the South China Sea Soft Coral Sarcophyton elegans"

_marinedrugs, 2022, doi:10.3390/md20090574_

Round 1

Reviewer 1 Report

The authors reported the identification of five new cembranes from the soft coral Sarcophyton elegans. The structures of the new isolates were elucidated using conventional NMR methods, and the absolute configurations were determined using the TDDFT-EDC calculation followed by comparison with experimental ECD values. In addition, anti-inflammatory activity of the natural compounds was evaluated employing a zebrafish model.

It seems that the relative configurations in compounds 2-5 had been unambiguously determined using NOESY experiments. However, there are some issues in the determination of C-3 configuration in compound 1. Although the C-3 configuration was determined by observing the 1D and 2D NOE signals between H-2 and H-3 to locate these protons on the same side, protons within three bonds can often give a NOE signal that causes misinterpretation. This argument can be supported by the 3D structures in figure 3 and figure S3 (1a in SI). If compound 1 actually exists like the major conformer represented in figure 3 and S3, NOESY correlations for H3-18/H-2 and H-2/H-3 cannot be observed because H3-18 and H-3 are located on the opposite side from H-2. Instead, NOESY signals for H-3/H3-18 and H-2/H-5 would be expected. In the 3D model of compound 1, H-2 and H-3 look like in an anti-relationship (180 degree) each other, but the experimental J value is only 4.2 Hz. This information also indicates that the major conformer provided in the manuscript and SI was improperly calculated or the structure assignment was imperfect. Considering the geometry of the 3-(E)-olefin in co-isolates (2-8), the epoxide of 1 may be in an anti-relationship between H-3 and H3-18 to give 2S*,3R*,4R*. In addition, I do not understand why the crucial references related to the isolation of 3,4-epoxy-cembranolides were missed in the manuscript, even these articles could provide criteria to determine the structure of compound 1 (Chem. Pharm. Bull. 2007, 55, 766.; Mar. Drugs, 2012, 10, 306.; J. Nat. Prod. 1992, 55, 1430.).

The manuscript also includes some minor mistakes that need to be modified:

Line 26: “belonging” to “belonging to”

Line 26~30: the sentence should be rewritten.

Line 37: “To pursuit of” to “To pursue”

Line 125: “CP4+” Are you sure that you conducted this experiment? Based on the information in the SI, it seems that the structure of 4 was determined by DP4+ calculation.

Line 126: “the orientation of the asymmetric center” to “the configuration”

Line 135~137: the sentence should be rewritten. There is some confusion in interpreting the HMBC correlations.

Line 139: “indicated” to “indicated that”

Line 151: “was” to “was used”

Reviewer 2 Report

The manuscript entitled “Sarcoeleganolides C–G, five New Cembranes from the South China Sea soft Coral Sarcophyton elegans” describes the discovery of five undescribed cembranes, sarcoeleganolides C–G, from the soft coral Sarcophyton elegans collected off the Yagong island, South China Sea. The structures were determined by MS, NMR analysis, along with chemical computations. A new compound 3 showed better anti-inflammatory activity than the positive control indomethacin at 20 uM in the zebrafish model. The structural elucidation is logical and the discovery of the new anti-inflammatory cembranes are pretty important for the field. I suggest that this manuscript may be suitable for publication in Marine Drugs after minor revision.

Here are some minor comments:

Abstract:

Yagong island into “Yagong Island”

QM-NMR and TDDFT-ECD calculations

Anti-inflammatory?

“And soft corals belonging…” into “Soft corals of the genus…”

“…a broad spectrum of biological activities…”

Please divide the first sentence (lines 25-30) into several shorter one.

“….multiple bioactivities…”

“…impressive achievements”? better rewrite this sentence. Also, should be “achievements have been made…”

“….Novel cembranoids…” better not say cembranoids novel. They are not.

Please re-write “These intriguing….. studies”.

“…led to the discovery of five new….”

Reviewer 3 Report

This paper describes the isolation and structures of five new cembranes isolated from marine soft corals. Their planar and stereochemistry were clearly established by exhaustive NMR analysis, and ECD calculation. Their characteristic structures include unsaturated carbon macrocycle with five-membered lactone, and several oxygenated functional groups including epoxides, hydroxy, and acetates. While a variety of structurally-related bioactive cembranes were already isolated, determination of relative and absolute stereochemistry of flexible carbon macrocycle structures that can adopt multiple conformations described in this paper is of great value. While the anti-inflammatory activity of five compounds against zebrafish in vivo were weak and cannot currently be assessed as of high value, this work might interest natural product chemists and medicinal chemists, and contribute to the development of new drug leads of marine origin. Thus, the reviewer recommends this paper for publication with the following considerations: 

1)    Stereochemistry at C11 hydroxy group of 5 in Figure 1 is difficult to understand. Dotted C-C bonds should be written on the obtuse side, not the acute side.

2)    Based on the dihedral angles in the calculated stereostructures of 1-5 in Figure 3, it should be evaluated whether the magnitudes of the coupling constants are reasonable. For example, H-2/H-3 in 1, H-5/H-6/H-7 in 3, H-5/H-6 in 4, and H-10/H-11 in 5. 

3)    There are a lot of typographical errors and logically inconsistent data are included in Table 1. For example, the J values of H-2 and H-3 in 1 and 5, H-5a/H-6, H-5b/H-6 in 3, H-6/H-7 in 4 should be identical, respectively. The J patterns “t,d” of H-11 in 1 and “d” of H-14a in 3 lacks J values, and “s; 5.02s” of H-20 in 5 is incorrect. Please carefully reanalyze the data and provide mature Table. 

4)    Zebrafish assay in Figure 5 is interesting, but a more detailed description of the experimental principles and interpretation of the results is required. How the green dots differ between control and CuSO4 treatment, and how the vertical axis of the graph changes? I cannot be evaluated just by looking at the photograph. There are little explanations as to what specific changes were observed in the treatment of the positive control indomethacin and the five compounds, and how they should be evaluated. 

5)    Copper sulfate is generally highly toxic and is expected to have severe effects on larval survival and normal development. Can administration of indomethacin or these compounds inherently avoid toxicity and lead to normal development? Otherwise, it would be misleading to conclude that there is a significant effect in vivo. Instead, carrageenan-treated mice for an in vivo inflammation model, or LPS-treated cultured macrophages to suppress inflammation are more common and appropriate for primary screening. The reviewer thinks that it is necessary to explain why this zebrafish fluorescent in vivo assay is better.

Round 2

Reviewer 1 Report

There is a minor revision.  

Line 140~141: "~correlations from H2-20 to C-11, C-12 and C-13 and H3-19 to C-7, C-8 and C-9." to "~correlations: from H2-20 to C-11, C-12, and C-13; from H3-19 to C-7, C-8, and C-9." 

Author Response

Q1: Line 140~141: "~correlations from H2-20 to C-11, C-12 and C-13 and H3-19 to C-7, C-8 and C-9." to "~correlations: from H2-20 to C-11, C-12, and C-13; from H3-19 to C-7, C-8, and C-9."

A1 for Q1: Line 140~141: "~correlations from H2-20 to C-11, C-12 and C-13 and H3-19 to C-7, C-8 and C-9." was revised to "~correlations: from H2-20 to C-11, C-12, and C-13; from H3-19 to C-7, C-8, and C-9."